# Understanding Cervical Cancer Screening Attendance: Barriers and Facilitators in a Representative Population Survey

**DOI:** 10.3390/cancers17040706

**Published:** 2025-02-19

**Authors:** Bogdan Florin Covaliu, Alina Ioana Forray, Mirela Tomic, Cătălin Vlad, Patriciu Achimaș Cadariu, Carmen Ungurean, Adriana Melnic

**Affiliations:** 1Department of Community Medicine, “Iuliu Hațieganu” University of Medicine and Pharmacy, 400012 Cluj-Napoca, Romania; 2Faculty of General Medicine, “Iuliu Hațieganu” University of Medicine and Pharmacy, 400012 Cluj-Napoca, Romania; mirela.tomic@publichealth.ro; 3The Oncology Institute “Prof. Dr. Ion Chiricuţă” Cluj Napoca, 400015 Cluj-Napoca, Romania; 4Department of Oncology, “Iuliu Hațieganu” University of Medicine and Pharmacy, 400012 Cluj-Napoca, Romania; 5National Institute of Public Health in Romania, 050463 Bucharest, Romaniaadriana.melnic@insp.gov.ro (A.M.); 6Romanian Cancer Society, 400090 Cluj-Napoca, Romania

**Keywords:** cervical cancer, cancer screening, secondary prevention, preventive healthcare services, participation, psychosocial factors, HPV, sexual health, early detection of cancer, Romania

## Abstract

The impact of various factors influencing cervical screening participation was analyzed in this nationally representative cross-sectional study. This work aims to assess socio-demographic characteristics, health practices, sexual history, and personal health views regarding organized cervical screening programs, as they are important mediators in screening engagement. Knowing the interplay between different factors could suggest the specifics of screening barriers in a population. This study might serve as a reference for creating adapted targeted interventions to meet the needs and bridge the existing identified gaps in the current national cervical screening program in Romania.

## 1. Introduction

Cervical screening, a critical preventive measure, is pivotal in the early detection and intervention of precancerous conditions within the cervix, ultimately aiming to reduce cervical cancer morbidity and mortality [1,2]. Organized cervical screening programs have substantially reduced the incidence and mortality rates of cervical cancer in various countries through regular and equitable screening [3,4,5]. However, disparities in screening uptake persist, particularly among socioeconomically deprived communities, with evidence suggesting lower participation rates among women from lower socioeconomic backgrounds compared to their higher socioeconomic counterparts [6,7,8,9]. The existing inequalities in cervical cancer screening pose a significant challenge, particularly due to the increased risk of mortality associated with non-attendance at routine screenings [4,10]. There are also various factors that affect a woman’s decision to participate in cervical screening, including psychological, sociocultural, and environmental factors [8,11].

Romania has the highest mortality rate for cervical cancer among all EU countries [12,13], with approximately 18.3 women per 100,000 dying every year from this disease, while the mortality rate for this type of cancer in the EU is 6.0 per 100,000 inhabitants on average [14,15]. Inequities in screening and Human Papillomavirus (HPV) vaccination programs further contribute to this country’s high cervical cancer incidence and mortality rates [16]. According to Eurostat, in 2021, the proportion of Romanian women aged 20 to 69 who had been screened for cervical cancer within the previous three years was the lowest among the EU Member States, at just 3.9%. Nearly half of the female population aged 20–69 years (47.4%) reported in 2019 that they had never had a cervical smear test [17]. The implementation of the cervical screening program in Romania faces several challenges, including poor procedures, low participation rates, and a lack of awareness campaigns [12,18,19]. The need for major awareness campaigns, functional screening programs, and coherent funding is emphasized [12,16].

To address this public health challenge, Romania launched a national, population-based cervical cancer screening program in 2012 [20]. This program offers free Babes–Papanicolaou (Pap smear) testing to women aged 25–64, which is a target population encompassing approximately 5.4 million women [21], aiming to detect precancerous lesions and reduce the incidence of invasive cervical cancer [22,23]. While the program is free of charge for eligible women, participation rates have remained strikingly low. In the program’s initial four years, only 16.9% of eligible women in the North-Eastern region of Romania were screened [20]. National coverage data indicate that no more than 10% of the target population benefits from screening [21], with participation being particularly low among vulnerable populations, such as Roma women [24].

Complementing the screening program, Romania has also implemented HPV vaccination initiatives. An initial HPV vaccination program was introduced between 2008 and 2010; however, this effort was unsuccessful, achieving coverage of less than 2% of the target population [25]. This low uptake has been attributed to various factors, including a lack of public awareness and trust in the vaccine. As of December 2023, the HPV vaccination has been reintroduced into the National Immunization Program. It is now freely available to both girls and boys aged 11–18, with partial reimbursement offered to women over 19 years of age [26]. The long-term impact of this renewed vaccination effort, particularly in light of the previous program’s challenges, remains to be evaluated.

The barriers to cervical screening in Romania are multifaceted, such as individual reasons, as the lack of time and financial resources were reported as key reasons for women postponing routine checks [12,16]. Other studies further emphasize systemic barriers, including complicated procedures and underfunding [20,27]. The influence of cultural and social norms on screening attendance is also significant, which underscores the role of socio-demographic factors, healthcare system issues, and lack of health literacy as additional barriers [28,29,30]. These findings collectively underscore the need for a comprehensive approach to address the barriers to cervical screening in Romania. Gaining insight into the traits of those who do not attend screenings is crucial in boosting participation rates.

The aim of this study is to identify the determinants that hinder participation in cervical cancer screening among Romanian women. This cross-sectional survey assesses the influence of socio-demographic factors, health behaviors, sexual history, and personal health beliefs on screening uptake. Targeting a nationally representative sample of women aged 25 to 64, this study seeks to pinpoint key barriers to inform targeted interventions within the national cervical cancer screening program, aiming to enhance public health outcomes through improved participation rates.

## 2. Materials and Methods

### 2.1. Study Design

This study employs a cross-sectional design to investigate the determinants of cervical cancer screening non-participation among Romanian women. Conducted as part of the broader initiative “Integration of Primary HPV Screening into the National Cervical Cancer Screening Program”, this research was collaboratively undertaken by the Prof. Dr. Ion Chiricuță Oncology Institute in Cluj-Napoca and the National Institute of Public Health in Romania. Data collection was outsourced to the Romanian Institute for Evaluation and Strategy (RIES), which utilized the Computer-Assisted Telephone Interviewing (CATI) methodology. All data were collected via telephone surveys conducted in February and March 2020. No incentives were offered for participation.

### 2.2. Participants

This study targeted screening-eligible women aged 25 to 64 years living in Romania. The sample was selected randomly from a nationally representative pool that covered all regions of Romania. Participants were included in the study if they met the following criteria: they were women aged between 25 and 64 years, residents of North-West, West, South-West, North-East, South-East, Central, South, and Bucharest-Ilfov, and had responded to the survey invitation. Furthermore, eligible participants had no confirmed diagnosis of cervical cancer and no history or evidence suggestive of cervical cancer pathology. Exclusion criteria were applied to ensure the integrity and focus of the study population. Women under 25 or over 64 years of age and non-residents of the specified region(s) were excluded. Individuals who did not complete the relevant sections of the survey necessary for this analysis were also excluded. In accordance with the national screening program guidelines [23], women with a congenital absence of the cervix, those who had undergone a total hysterectomy for benign conditions, or those with a confirmed diagnosis of cervical cancer or other forms of genital cancer were not eligible for participation.

To ensure a maximum margin of error of ±2.5% at a 95% confidence level, a random sample of 1605 participants was chosen using a probabilistic sampling method. Sampling weights provided by IRES ensured data representativeness for age, social grade, and region. Women who did not provide enough information to determine their screening status were excluded from the current analysis (N = 49). This report analyzes the selected sample of 1556 Romanian participants, comparing those who reported never having been screened for cervical cancer (non-participants) with those who reported at least one prior screening (participants) to identify factors associated with non-participation.

### 2.3. Ethical Considerations

The Ethics Committee for the Institute of Oncology, “Prof. Dr. Ion Chiricuță” in Cluj-Napoca, reviewed and approved the study. Before responding to the survey, all study participants provided informed consent verbally and were informed that they could withdraw their consent at any time during the data collection process.

### 2.4. Data and Measurement

The questionnaire employed in this study was adapted and refined based on previous research conducted by Andreassen et al. [28]. This was performed through a comprehensive collaboration between specialists at the “Prof. Dr. Ion Chiricuță” Oncology Institute and the National Institute of Public Health in Romania. The questionnaire encompasses 49 questions, predominantly employing a closed-ended format with predefined categorical response options and open-answer options (Appendix A). The estimated completion time for the questionnaire was between 30 and 35 min.

### 2.5. Dependent Variable: Cervical Cancer Screening Non-Participation

The primary outcome variable in this study is lifetime non-participation in cervical cancer screening. This was defined as a binary variable (yes/no) based on responses to question Q8_2 (Appendix A): “Have you ever taken the Papanicolaou test?” Respondents answering ‘no’ were categorized as non-participants, while those answering with any option indicating prior testing were categorized as participants.

### 2.6. Explanatory Variables

For the purposes of this specific manuscript, a subset of questions was selected for analysis, focusing on factors hypothesized to be most directly relevant to cervical cancer screening non-participation. The questionnaire was structured into several sections, with the following questions selected for inclusion in the present analysis:

The socio-demographic section included questions capturing information on age (Question P0), ethnicity (Question P1), educational attainment (Question P3), occupation (Question P4), income (Question P5), membership in a socially disadvantaged group (Question Q14), and residence location (Question P6) [1,6,7,31,32,33]. For analysis, age was categorized into three groups: ≤30 years, 31–49 years, and ≥50 years. Ethnicity was categorized as “Romanian” and “Non-Romanian”. Educational attainment was classified as “Below Secondary”, “Secondary”, and “Tertiary”. Occupation was categorized as “Employed” (including self-employed and partially employed), “Unemployed/Inactive” (including housewives and jobless individuals), and “Other” (retirees, students, and those receiving social aid). Income was categorized into three brackets: less than RON 1000 (approximately EUR 200), RON 1000 to 3000 (EUR 200–600), and over RON 3000 (over EUR 600).

The health-related section comprised four questions pertaining to the individual’s perception of their health status (Q1), the existence of long-term health conditions (Q2), current tobacco use (Q2_1), and the frequency of visits to a primary care physician (Q3_1) [9,33,34,35]. Self-rated health status was categorized as Good, Neutral, and Bad for analysis.

The section on sexual behavior [11,36,37] comprised four questions regarding the use of contraceptives (Q4_3), the number of childbirths (Q4_4_1, Q4_4_1i), the age of initial sexual activity (Q4), and the total number of sexual partners over one’s lifetime (Q4_2). Age at sexual debut was categorized as follows for analysis: after 25, 19–24, and less than 18. The lifetime number of partners was categorized as ≥4 partners and 1–3 partners. The number of births was classified as 0, 1, and ≥2 births.

The “Cervical cancer awareness, healthcare engagement, personal perceptions, and beliefs” section [38,39,40,41,42] included 15 yes/no questions that assessed factors potentially influencing cervical cancer screening participation. Specifically, questions explored whether participants had heard of a cervical cancer test (Question Q6), had received information about screening from their primary care physician or nurse (Question Q7), had heard of the Human Papillomavirus (Question Q9), and were aware of the National Screening Program (Question Q11). Healthcare engagement was assessed through questions about referrals to OB/GYN specialists and offers to perform the screening test (Question Q7). Participants’ attitudes and beliefs were assessed through questions (Question Q10) probing trust in medical services, beliefs about prevention and curability, perceptions of risk and cost, and attitudes toward women with cervical cancer. Reasons for not participating in cervical cancer screening were elicited through a question with predefined options for those who had never been screened (Question Q8_2_4).

#### Statistical Analysis

In our initial analytical phase, we employed descriptive statistics to delineate the distribution of characteristics among participants, categorizing them by their screening status. Subsequently, we evaluated the impact of these characteristics on non-participation through logistic regression analyses, both unadjusted and adjusted for potential confounders, such as age, education, employment status, and income, as socio-demographic factors have been demonstrated to influence screening behaviors. Finally, the relationships between independent variables and non-participation in cervical cancer screening were quantified using odds ratios, presented with 95% confidence intervals. All statistical analyses were conducted using SPSS Software (version 29; MacOS). We set the significance threshold at a *p*-value of less than 0.05.

## 3. Results

### 3.1. Demographic Profile and Associations with Non-Participation

The study sample comprised 1556 women aged 25–64 years. Screening histories varied significantly: 25.1% of women had never been screened for cervical cancer, 15.9% had been screened once, and 58.9% had been screened at least twice. Table 1 presents the socio-demographic characteristics of the sample and their associations with cervical cancer screening non-participation. The current sample displayed a diverse age range with a mean age of 46.0 years (SD = 10.63). The majority of participants were Romanian (93.0%). Educational attainment varied, with 49.3% having secondary education and a concerning 7.3% having below secondary education. Regarding employment, 64.4% were employed or self-employed, 17.4% were unemployed/inactive, and 18.1% were in other situations (retired, students, etc.). Income disparities were evident, with the majority earning between RON 1000 and 3000. A substantial proportion (38.1%) reported belonging to a disadvantaged group. The sample’s even distribution of urban (50.7%) and rural (49.3%) participants indicates diverse geographical access challenges.

As shown in Table 1, several socio-demographic factors were significantly associated with non-participation in cervical cancer screening. Women aged 30–49 had a significantly lower non-participation rate (20.5%) compared to those aged ≤30 (28.7%) and ≥50 (30.1%), with an unadjusted odds ratio (uOR) of 0.64 (95% CI: 0.43–0.96). Educational attainment showed a strong association, with non-participation rates decreasing significantly with increasing education: 47.3% for below secondary (uOR: 2.78, 95% CI: 2.13–3.65), 31.2% for secondary (uOR: 5.52, 95% CI: 3.57–8.53), and 14.0% for tertiary education. Employment status significantly influenced screening attendance; employed or self-employed women had lower odds of non-participation compared to unemployed or inactive women (uOR: 2.15, 95% CI: 1.59–2.90). Similarly, income levels were critically linked to screening behaviors; women earning less than RON 1000 exhibited the highest odds of non-participation (uOR: 4.20, 95% CI: 2.85–6.19). Rural residents had a significantly higher non-participation rate (32.7%) than urban residents (17.7%) (uOR: 2.25, 95% CI: 1.77–2.85). Women belonging to a disadvantaged group also had a higher rate of non-screening (31.7%) compared with those who did not belong to a disadvantaged group (21.1%).

### 3.2. General Health, Lifestyle Characteristics, Sexual Behavior, and Associations with Non-Participation

Table 2 presents the overall prevalence of health-related and lifestyle characteristics and sexual behavior and the associations between health-related and lifestyle characteristics, sexual behavior, and non-participation in cervical cancer screening. Overall, the majority of the sample (86.8%) rated their health as “good”, while 70.5% reported no chronic diseases. A considerable proportion of the sample (24.4%) reported being current smokers. Only 56.4% reported visiting their family physician more than twice per year. Regarding sexual behavior, while the majority (80.8%) reported only 1–3 lifetime partners, a notable 13.0% reported four or more partners. A large proportion (83.3%) reported not using any contraceptive method.

Women reporting poor health had a non-participation rate of 33.3%, compared to 24.4% for those reporting good health and 24.4% for neutral in the unadjusted analysis with a uOR of 1.54 (95% CI: 1.04–2.29) (Table 2). Women visiting their family physician less than once per year had a significantly higher non-participation rate (30.8%) compared to those visiting more than twice per year (22.3%), demonstrated by a uOR of 1.54 (95% CI: 1.16–2.04). The number of lifetime sexual partners also played a role; women with 1–3 partners had elevated odds of non-participation (uOR: 2.05, 95% CI: 1.35–3.10), and this trend was particularly strong among women with multiple births, where those having two or more births exhibited a uOR of 2.10 (95% CI: 1.44–3.08).

Adjusted odds ratios (aOR), controlling for demographic factors such as age, education, employment status, and income, revealed that chronic diseases slightly reduced the odds of non-participation (aOR: 0.72, 95% CI: 0.54–0.98). Furthermore, women visiting their family physician less than once yearly showed significantly higher odds of non-participation (aOR: 1.97, 95% CI: 1.43–2.70). The relationship between sexual debut and screening engagement also remained significant, with those initiating sexual activity between ages 19–24 having lower adjusted odds of non-participation (aOR: 0.51, 95% CI: 0.29–0.89).

### 3.3. Cervical Cancer Screening Awareness, Healthcare Engagement, Personal Perceptions and Beliefs, and Associations with Non-Participation

Table 3 and Table 4 present the associations between awareness, healthcare engagement, personal perceptions and beliefs, and non-participation. Table 3 shows that the overall awareness of key aspects of cervical cancer prevention was concerningly low. Alarmingly, only 48.8% of women had heard of the Human Papillomavirus. While 63.1% had heard of a test for cervical cancer, and only 51.0% were aware of the National Screening Program. Furthermore, only 22.1% reported receiving information on screening from their primary care physician or nurse, and only 25.2% had been referred to an OB/GYN specialist. Only 16.5% of women reported that the test was offered by the physician or nurse.

In the univariate analysis of factors influencing cervical cancer screening participation among Romanian women, several key elements related to awareness and healthcare engagement were significant (Table 3). A significantly higher proportion of women who had not heard of HPV (36.0%) were non-participants in cervical cancer screening compared to those who had heard of HPV (13.7%). Notably, the lack of awareness about HPV was associated with the highest odds of non-participation (uOR: 3.53, 95% CI: 2.74–4.55). Women who had not received a referral to an OB/GYN specialist from their primary care provider had a substantially higher non-participation rate (29.0%) than those who had received a referral (13.3%) (uOR: 2.66, 95% CI: 1.96–3.66). Similarly, women whose physicians did not offer the screening test showed elevated odds of non-participation (uOR: 2.30, 95% CI: 1.44–3.66). Awareness of the cervical cancer test and the national screening program were also crucial, showing increased likelihoods of screening participation with uORs of 1.55 (95% CI: 1.22–1.97) and 1.69 (95% CI: 1.34–2.13), respectively.

In the multivariable analysis, after adjusting for confounding factors such as age, education, employment status, and income, the associations remained largely significant, albeit slightly attenuated. The adjusted odds for a lack of HPV awareness remained notably high (aOR: 2.45, 95% CI: 1.85–3.25), indicating a persistent gap in essential health knowledge impacting screening behavior. The effects of not receiving a referral to a specialist (aOR: 2.96, 95% CI: 2.09–4.19) and not being offered a screening test (aOR: 2.24, 95% CI: 1.38–3.63) continued to show strong associations with increased odds of non-participation.

Table 4 highlights prevalent perceptions and beliefs. While a large majority (80.8%) reported trusting local medical services, a concerning 22.2% perceived the cost of a Pap smear test as expensive. Notably, 7.8% of women reported avoiding women diagnosed with cervical cancer, indicating a significant level of social stigma. Furthermore, a small minority (9.4%) believed that intimate hygiene could prevent cervical cancer. The majority (89.3%) believed in the curability of cervical cancer.

Notably, the perception of the cost of a Pap smear test as expensive was a major deterrent; 27.5% of women who perceived the test to be expensive were non-participants compared to 15.0% of those who did not perceive it to be expensive (uOR: 2.15, 95% CI: 1.60–2.89). A significantly higher proportion of women who reported avoiding those diagnosed with cervical cancer were non-participants (49.6%) compared to those who did not report such avoidance (22.6%) (uOR: 3.43, 95% CI: 2.35–5.00). Additionally, viewing bleeding between menstruations as normal was linked to increased non-participation (uOR: 1.50, 95% CI: 1.00–2.25). Conversely, the belief that intimate hygiene cannot prevent cancer demonstrated a protective effect, with women holding this belief more likely to participate in screening (uOR: 0.60, 95% CI: 0.39–0.94).

In the multivariable model, significant associations remained after adjusting for age, education, employment status, and income, but some were attenuated. The perceived high costs of Pap smear tests continued to be a significant barrier (aOR: 1.76, 95% CI: 1.27–2.43), underscoring the impact of financial considerations on health behaviors. Similarly, the stigma associated with cervical cancer diagnosis remained a significant factor, although the odds were slightly lower (aOR: 2.28, 95% CI: 1.50–3.45).

### 3.4. Barriers to Participating in Cervical Cancer Screening

In our analysis of barriers to cervical cancer screening among 391 unscreened Romanian women (Figure 1), the most commonly reported reasons were the absence of symptoms, accounting for 32.7%, and a lack of awareness regarding the necessity of screening, reported by 10.7% of the participants. Negligence or hesitation was cited by 10.2%, while 9.5% indicated a lack of time as a deterrent. Beliefs of being healthy and, therefore, not needing the test were expressed by 6.2% of the women. Additional barriers included a lack of recommendation from healthcare professionals, mentioned by 5.1%, and financial constraints, with 4.6% reporting insufficient funds to undergo screening. Fear-related factors such as apprehension about visiting doctors and fear of receiving a cancer diagnosis were noted by 3.9% and 2.8%, respectively.

## 4. Discussion

Despite the pivotal role of cervical screening in detecting early cancerous lesions, Romania still ranks last among the EU member states for cervical cancer screening participation. Health inequalities and existing gaps in the screening process may contribute to low rates of engagement [12,13]. Identifying the most influential factors for different subpopulations can help tailor effective interventions addressing the current gap and limited research in Romania on this topic. Thus, the present study was designed to explore the facilitators and barriers to cervical cancer screening, aiming to assess the influence of socio-demographic factors, health behaviors, sexual history, and personal health beliefs on cervical screening uptake.

Our study’s findings align with previous research, indicating that socio-demographic variables such as age, education, employment, income, and urban residency significantly influence participation in cervical cancer screening. Likewise, in another study, educational attainment and financial situations were shown to be substantially correlated with screening adherence, low family income, and education level, which were linked with non-attendance [7,43]. The favorable correlation between screening uptake and educational attainment was also sustained, considering health literacy as a mediator, as the response to general population-based health education initiatives is restricted by low literacy [31]. Also, several studies [44,45] have consistently demonstrated that rural residents are less likely to participate in cervical cancer screening compared to their urban counterparts, highlighting significant disparities in healthcare access and utilization between these populations. Health status also represents an important predictor of cervical screening engagement. According to our analysis, women reporting poor health had higher odds of non-participation in screening. These results are in agreement with previous data, as women with comorbidities are less likely to engage in cervical cancer screening because they tend to concentrate on their already present medical illness to the disadvantage of preventive services [34]. Notably, a poor health status suggested by a high BMI and older age were significant predictors of non-participation in cervical cancer screening [46]. Sexual health factors such as multiple births and several sexual partners impacted cervical screening. Contrary to our results, multiparity was linked to higher screening service utilization [36]. Thus, acknowledging the influence of socio-demographic factors on cervical screening participation is crucial because it can guarantee equitable access to preventive care, prioritizing groups underutilizing these services [9].

A significant finding is the association between female family doctor visits and screening involvement. These findings support the work of other papers linking GP visits with screening engagement [47], highlighting the role of primary care in promoting cervical cancer screening through increased GP involvement that could potentially enhance screening uptake [48]. Moreover, physician recommendation is among the strongest predictors of cervical cancer screening [38,41]. This is consistent with our results, as the lack of referral from a GP showed strong associations with increased odds of female non-participation in screening. This observation may support the idea of GPs shifting their roles from gatekeeping toward becoming active stakeholders. More precisely, compared to suggesting a Pap smear performed by a doctor, directly providing GPs with self-sampling kits could lead to a considerable increase in cervical cancer screening participation rates [40]. Additionally, in the UK, GPs can access Clinical Cancer Decision Tools, which help detect patients who should be screened, as they inform family doctors about the warning signs and symptoms of cancer so they can examine and refer patients promptly [49].

The most relevant result from a public health perspective that stands out from the results reported earlier is the lack of awareness about HPV and the existing national screening programs, which remained notably high after adjusting for confounding factors. Despite the high level of education of females in our sample, the lack of awareness was a barrier to cervical cancer screening, indicating the urgent need for delivering targeted and effective interventions to bridge the knowledge–practice gap. Nevertheless, a study suggested that higher education levels are generally associated with higher rates of adherence to cervical screening guidelines, considering their greater access to healthcare information, ability to navigate healthcare systems, and proactive health-seeking behaviors [50]. Previous studies on Romanian women have underscored the necessity for improvements in service provision and enhancements in information and counseling within obstetrics–gynecology settings [51].

One interesting finding is that a vast majority of the sample believe that intimate hygiene can prevent cancer, with non-participants being more likely to believe that intimate hygiene can prevent cancer than participants. It is important to note that the role of intimate hygiene in preventing cervical cancer is not well established. While some studies have suggested that maintaining proper intimate hygiene, including menstrual hygiene, can play a role in preventing cervical cancer [52,53,54], the primary prevention methods for this disease are HPV vaccination and early detection through screening [55]. However, there is a need for improved knowledge and practice of genital hygiene, especially among healthcare workers [56].

The most reported reasons for not participating in screening were the belief that an absence of symptoms meant there was no need for screening, followed by a lack of awareness regarding the necessity of screening, which were the most commonly reported reasons for non-participation. These barriers were also identified as major impediments, as many women were unaware of the importance of screening without symptoms [57]. Thus, our results, in line with previous supporting studies in the literature, emphasize the need for increased education and awareness campaigns to highlight the importance of regular cervical screening, even in the absence of symptoms, to improve participation rates and early detection.

### Limitations

This study has several limitations that should be considered when interpreting the findings. Firstly, our reliance on self-reported data for screening history, sexual behavior, and personal perceptions introduced the potential for recall bias and social desirability bias. Participants may not have accurately remembered past screening events or may have over-reported socially desirable behaviors. Secondly, the “lifetime perspective” on screening history, while providing information on those who had never been screened, presents interpretative challenges, as the opportunity for screening differs significantly between younger and older women. Thirdly, the cross-sectional design of this study precludes establishing causal relationships between the identified factors and screening non-participation; thus, we could only identify associations. Fourthly, the use of Computer-Assisted Telephone Interviewing (CATI), while efficient, may have introduced selection bias by excluding individuals without access to telephones or those less likely to answer calls from unknown numbers. The data collected are also from early 2020, and the situation and screening rates could have changed since then. Finally, although we tried to include people from all regions, the study population may present some geographical limitations. To address these limitations in future research, longitudinal studies that track screening behavior over time would be beneficial. Whenever possible, linking survey data to objective screening records from medical registries would mitigate recall bias. Mixed methods approaches, incorporating qualitative interviews to explore the nuances of screening decisions, could provide richer insights. Furthermore, future studies should consider employing sampling techniques designed to maximize the inclusion of underrepresented and harder-to-reach populations, potentially through oversampling or targeted outreach efforts. Finally, focusing on screening participation within nationally recommended intervals or shorter timeframes (e.g., the past 1–2 years) would provide a more accurate and age-relevant assessment of current screening adherence.

## 5. Conclusions

Although cervical screening has a significant role in the early detection of cancerous lesions, women have low participation rates because of a lack of awareness and absence of symptoms. Socio-demographic characteristics, general health, sexual behaviors, personal beliefs, and cervical screening awareness are significant predictors of participation. Our study is of significant importance for the public health field as it highlights the influence of various predictors on cervical screening participation, which should remain at the fore of future interventions for raising cervical cancer awareness and contributing to higher screening rates.

## Figures and Tables

**Figure 1 cancers-17-00706-f001:**
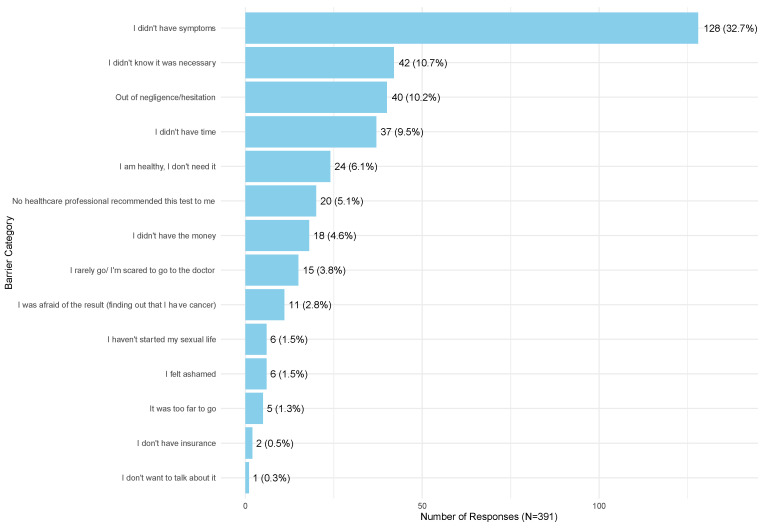
Unscreened women’s barriers to participating in cervical screening.

**Table 1 cancers-17-00706-t001:** Socio-demographic characteristics of 1556 women aged 25–64 years from the general Romanian female population and association with non-participation in cervical cancer screening.

Characteristic	Total N (%)	Participating in Cervical Cancer Screening		Crude OR (95% CI)
		No N (%)	Yes N (%)	
**Age**				
≤30 years	143 (9.2)	41 (28.7)	102 (71.3)	Ref.
30–49 years	782 (50.3)	160 (20.5)	622 (79.5)	0.64 (0.43–0.96) ***
≥50 years	631 (40.6)	190 (30.1)	441 (69.9)	1.07 (0.72–1.60)
**Ethnicity**				
Non-Romanian	106 (7.0)	33 (31.1)	73 (68.9)	Ref.
Romanian	1411 (93.0)	343 (24.3)	1068 (75.7)	0.71 (0.46–1.09)
**Educational attainment**				
Tertiary	659 (43.4)	92 (14.0)	567 (86.0)	Ref.
Secondary	748 (49.3)	233 (31.1)	515 (68.9)	2.78 (2.13–3.65) ***
Below Secondary	110 (7.3)	52 (47.3)	58 (52.7)	5.52 (3.57–8.53) ***
**Occupation**				
Employed/Self-employed	957 (64.4)	190 (19.9)	767 (80.1)	Ref.
Unemployed/Inactive	259 (17.4)	90 (34.7)	169 (65.3)	2.15 (1.59–2.90) ***
Other	269 (18.1)	92 (34.2)	177 (65.8)	2.09 (1.55–2.82) ***
**Monthly net personal income**				
≥ RON 3000	375 (25.3)	43 (11.5)	332 88.5)	Ref.
RON 1000–3000	766 (51.6)	208 (27.2)	558 (72.8)	2.87 (2.01–4.10) ***
< RON 1000 or None	343 (23.1)	121 (35.3)	222 (64.7)	4.20 (2.85–6.19) ***
**Place of residence**				
Urban	789 (50.7)	140 (17.7)	649 (82.3)	Ref.
Rural	767 (49.3)	251 (32.7)	516 (67.3)	2.25 (1.77–2.85) ***
**Disadvantaged group**				
No	963 (61.9)	203 (21.1)	760 (78.9)	Ref.
Yes	593 (38.1)	188 (31.7)	405 (68.3)	1.73 (1.37–2.19) ***
**Total**	**1556 (100.0)**	**391 (25.1)**	**1165 (74.9)**	

Note. *** *p* < 0.001. CI, confidence interval; OR, odds ratio. Bold text indicates main variables in the analysis. Ref. indicates the reference category in the logistic regression analysis.

**Table 2 cancers-17-00706-t002:** Associations between health-related, lifestyle, and sexual behavior characteristics and non-participation in cervical cancer screening among 1556 women aged 25–64 years from the general Romanian female population.

Characteristic	Total N (%)	Participating in Cervical Cancer Screening		Crude OR (95% CI)	Adjusted OR ^a^ (95% CI)
		No N (%)	Yes N (%)		
**Self-rated health status**					
Good	1351 (86.8)	330 (24.4)	1021 (75.6)	Ref.	Ref.
Neutral	78 (5.0)	19 (24.4)	59 (75.6)	0.98 (0.58–1.69)	0.57 (0.31–1.02)
Bad	123 (7.9)	41 (33.3)	82 (67.7)	1.54 (1.04–2.29) *	0.93 (0.60–1.45)
**Chronic disease**					
No	1095 (70.4)	276 (25.2)	819 (74.8)	Ref.	Ref.
Yes	459 (29.5)	115 (25.1)	343 (74.9)	1.00 (0.78–1.29)	0.72 (0.54–0.98) *
**Current smoking**					
No	1177 (75.6)	294 (25.0)	883 (75.0)	Ref.	Ref.
Yes	379 (24.4)	97 (25.6)	282 (74.4)	1.03 (0.79–1.34)	1.08 (0.81–1.44)
**Family physician visit frequency**					
More than 2 times per year	878 (56.4)	196 (22.3)	682 (77.7)	Ref.	Ref.
Once per year	317 (20.4)	84 (26.5)	233 (73.5)	1.25 (0.93–1.68)	1.54 (1.11–2.14) *
Less than once per year	338 (21.7)	104 (30.8)	234 (69.2)	1.54 (1.16–2.04) **	1.97 (1.43–2.70) ***
**Age at sexual debut**					
After 25	95 (6.1)	27 (28.4)	68 (71.6)	Ref.	Ref.
19–24	1024 (65.8)	243 (23.7)	781 (76.3)	0.78 (0.49–1.25)	0.51 (0.29–0.89) *
Less than 18	380 (24.4)	93 (24.5)	287 (75.5)	0.81 (0.49–1.35)	0.64 (0.39–1.05)
**Lifetime number of partners**					
≥4 partners	202 (13.0)	29 (14.4)	173 (85.6)	Ref.	Ref.
1–3 partners	1258 (80.8)	322 (25.6)	936 (74.4)	2.05 (1.35–3.10) **	1.31 (0.84–2.05)
**Number of births**					
0 births	255 (16.4)	40 (15.7)	215 (84.3)	Ref.	Ref.
1 birth	436 (28.0)	100 (22.9)	336 (77.1)	1.60 (1.06–2.39) *	1.22 (0.79–1.91)
≥2 births	625 (40.2)	176 (28.2)	449 (71.8)	2.10 (1.44–3.08) ***	1.36 (0.88–2.08)
**Contraceptive use**					
Yes	259 (16.6)	51 (19.7)	208 (80.3)	Ref.	Ref.
No	1296 (83.3)	339 (26.2)	957 (73.8)	1.44 (1.03–2.01) *	1.12 (0.76–1.64)

Note. * *p* < 0.05, ** *p* < 0.01, *** *p* < 0.001. CI, confidence interval; OR, odds ratio. ^a^ Adjusted for age, education, employment status, and income. Percentages in the ‘Total’ column may not sum to 100% due to missing values and/or rounding. Bold text indicates main variables in the analysis. Ref. indicates the reference category in the logistic regression analysis.

**Table 3 cancers-17-00706-t003:** Associations between participant awareness and primary care engagement with cervical cancer screening and non-participation in cervical cancer screening among 1556 women aged 25–64 years from the general Romanian female population.

Characteristic	Total N (%)	Participating in Cervical Cancer Screening		Crude OR (95% CI)	Adjusted OR ^a^ (95% CI)
		No N (%)	Yes N (%)		
**Heard about a test for the detection of cervical cancer**					
Yes	982 (63.1)	212 (21.6)	770 (78.4)	Ref.	Ref.
No	535 (34.3)	160 (29.9)	375 (70.1)	1.55 (1.22–1.97) ***	1.26 (0.97–1.64)
**Received information on screening from primary care physician/nurse**					
Yes	344 (22.1)	68 (19.8)	276 (80.2)	Ref.	Ref.
No	1193 (76.7)	315 (26.4)	878 (73.6)	1.45 (1.08–1.95) *	1.50 (1.09–2.07) *
**Primary care physician/nurse made a referral to OB/GYN specialist**					
Yes	392 (25.2)	52 (13.3)	340 (86.7)	Ref.	Ref.
No	1147 (73.7)	332 (29.0)	815 (71.0)	2.66 (1.96–3.66) ***	2.96 (2.09–4.19) ***
**Primary care physician/nurse offered to do screening test**					
Yes	256 (16.5)	59 (23.0)	197 (77.0)	Ref.	Ref.
No	1303 (83.7)	328 (25.2)	975 (74.8)	2.30 (1.44–3.66) ***	2.24 (1.38–3.63) **
**Received brochure with information regarding screening from primary care physician/nurse**					
Yes	231 (14.8)	38 (16.5)	193 (83.5)	Ref.	Ref.
No	1305 (85.2)	346 (26.5)	959 (73.5)	1.83 (1.26–2.65) **	1.99 (1.34–2.95) ***
**Heard of the Human Papillomavirus**					
Yes	759 (48.8)	104 (13.7)	655 (86.3)	Ref.	Ref.
No	795 (51.1)	286 (36.0)	509 (64.0)	3.53 (2.74–4.55) ***	2.45 (1.85–3.25) ***
**Heard of the National Screening Program for Cervical Cancer**					
Yes	794 (51.0)	162 (20.4)	632 (79.6)	Ref.	Ref.
No	754 (48.5)	228 (30.2)	526 (69.8)	1.69 (1.34–2.13) ***	1.47 (1.14–1.90) *

Note. * *p* < 0.05, ** *p* < 0.01, *** *p* < 0.001. CI, confidence interval; OB/GYN, obstetrics and gynecology; OR, odds ratio. ^a^ Adjusted for age, education, employment status, and income. Percentages in the ‘Total’ column may not sum to 100% due to missing values and/or rounding. Bold text indicates main variables in the analysis. Ref. indicates the reference category in the logistic regression analysis.

**Table 4 cancers-17-00706-t004:** Associations between personal perceptions and beliefs and non-participation in cervical cancer screening among 1556 women aged 25–64 years from the general Romanian female population.

Characteristic	Total N (%)	Participating in Cervical Cancer Screening		Crude OR (95% CI)	Adjusted OR ^a^ (95% CI)
		No N (%)	Yes N (%)		
**Trust in local medical services**					
Yes	1257 (80.8)	322 (25.6)	935 (74.4)	Ref.	Ref.
No	282 (18.1)	66 (23.4)	216 (76.6)	0.88 (0.65–1.20)	1.13 (0.81–1.57)
**Belief that intimate hygiene can prevent cancer**					
Yes	1388 (89.2)	362 (26.1)	1026 (73.9)	Ref.	Ref.
No	147 (9.4)	26 (17.7)	121 (82.3)	0.60 (0.39–0.94) *	0.92 (0.57–1.49)
**Belief in curability of cervical cancer**					
Yes	1389 (89.3)	341 (24.6)	1048 (75.4)	Ref.	Ref.
No	99 (6.4)	29 (29.3)	70 (70.7)	1.27 (0.81–1.99)	1.36 (0.83–2.21)
**Perception of bleeding between menstruations as normal**					
No	1406 (90.4)	337 (24.0)	1069 (76.0)	Ref.	Ref.
Yes	118 (7.6)	38 (32.2)	80 (67.8)	1.50 (1.00–2.25) *	1.21 (0.78–1.87)
**Perception of the cost of Pap smear test as expensive**					
No	956 (61.4)	143 (15.0)	813 (85.0)	Ref.	Ref.
Yes	346 (22.2)	95 (27.5)	251 (72.5)	2.15 (1.60–2.89) ***	1.76 (1.27–2.43) ***
**Perception of low personal risk of getting cervical cancer**					
No	820 (52.7)	193 (23.5)	627 (76.5)	Ref.	Ref.
Yes	436 (28.0)	117 (26.8)	319 (73.2)	1.19 (0.91–1.55)	1.23 (0.92–1.63)
**Avoidance of women diagnosed with cervical cancer**					
No	1405 (90.3)	317 (22.6)	1088 (77.4)	Ref.	Ref.
Yes	122 (7.8)	61 (50.0)	61 (50.0)	3.43 (2.35–5.00) ***	2.28 (1.50–3.45) ***
**Belief in need for regular gynecological check-ups**					
Yes	1520 (97.7)	378 (24.9)	1142 (75.1)	Ref.	Ref.
No	27 (1.7)	10 (37.0)	17 (63.0)	1.77 (0.80–3.91)	1.70 (0.72–4.01)
**Confidence in access to treatment if diagnosed**					
Yes	1449 (93.1)	358 (24.7)	1091 (75.3)	Ref.	Ref.
No	47 (3.0)	15 (31.9)	32 (68.1)	1.42 (0.76–2.66)	1.35 (0.68–2.66)

Note. * *p* < 0.05, *** *p* < 0.001. CI, confidence interval; OR, odds ratio. ^a^ Adjusted for age, education, employment status, and income. Percentages in the ‘Total’ column may not sum to 100% due to missing values and/or rounding. Bold text indicates main variables in the analysis. Ref. indicates the reference category in the logistic regression analysis.

## Data Availability

The dataset underpinning this study’s conclusions is accessible through the “Prof. Dr. Ion Chiricuță” Oncology Institute. The data can be obtained from the corresponding author upon reasonable request and with approval from the “Prof. Dr. Ion Chiricuță” Oncology Institute.

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
