# Peer review of "Understanding Cervical Cancer Screening Attendance: Barriers and Facilitators in a Representative Population Survey"

_cancers, 2025, doi:10.3390/cancers17040706_

Round 1

Reviewer 1 Report

Comments and Suggestions for Authors

Thank you for the opportunity to review manuscript ID: cancers-3472370. This manuscript aimed to investigate the factors limiting Romanian women's participation in cervical cancer screening.   

Minor comments:  

Line 78: Add a new paragraph in which the screening program for cervical cancer in women in Romania should be described (since when was screening introduced, for which age of women, which screening test is applied, whether it is free or paid), citing appropriate references. Also, present data on HPV vaccination in Romania (since when was HPV vaccination introduced, for which age of women and/or men, whether it is free or paid, what is HPV vaccination coverage), citing appropriate references.

Lines 107-117: Specify the inclusion and exclusion criteria in this study.

Line 365: Add a new paragraph to state and discuss the limitations of this work. Discuss the possibilities for eliminating the limitations of this work. 

Author Response

Comment 1: "Line 78: Add a new paragraph in which the screening program for cervical cancer in women in Romania should be described (since when was screening introduced, for which age of women, which screening test is applied, whether it is free or paid), citing appropriate references. Also, present data on HPV vaccination in Romania (since when was HPV vaccination introduced, for which age of women and/or men, whether it is free or paid, what is HPV vaccination coverage), citing appropriate references."
Response 1: We have revised the Introduction to integrate information about the Romanian cervical cancer screening program and the history and current status of HPV vaccination. The national screening program, launched in 2012, offers free Pap smear testing to women aged 25-64 (Nyanchoka et al., 2022; Păvăleanu et al., 2018; National Health Insurance House, 2012). We added details regarding a previous, unsuccessful HPV vaccination program (2008-2010) with very low coverage (European Commission, 2023), and the program's reintroduction in December 2023, which provides free vaccination for girls and boys aged 11-18 and partial reimbursement for women over 19 (UNICEF, 2023). This comprehensive overview of both prevention strategies provides a more complete context for our study.

Comment 2: "Lines 107-117: Specify the inclusion and exclusion criteria in this study."
Response 2: We have revised the Methods section to explicitly state the inclusion and exclusion criteria for participants in this study. We clarified that our study population was drawn from individuals eligible for the national cervical cancer screening program, thus mirroring the program's basic eligibility requirements (women aged 25-64, no prior cervical cancer diagnosis, and no congenital absence of the cervix or total hysterectomy for benign conditions) (National Health Insurance House, 2012). We also have further specified any additional inclusion or exclusion criteria applied within our study sample, such as requiring participants to be residents of a specific region or having responded to the survey invitation, to ensure the clarity and reproducibility of our research.

Comment 3: "Line 365: Add a new paragraph to state and discuss the limitations of this work. Discuss the possibilities for eliminating the limitations of this work."
Response 3: We have added a new paragraph to address the limitations of our study explicitly. This paragraph acknowledges the potential for recall and social desirability bias due to our reliance on self-reported data, the limitations of our cross-sectional design in establishing causality, and the potential selection bias introduced by the CATI methodology. We also discuss the limitations of a 'lifetime perspective' on screening history. To mitigate these limitations in future research, we suggest longitudinal studies, incorporating objective screening records, using mixed-methods approaches (combining quantitative surveys with qualitative interviews), and employing sampling strategies that maximize the inclusion of diverse populations, potentially including oversampling of harder-to-reach groups.

Reviewer 2 Report

Comments and Suggestions for Authors

The study design and the method of presenting the results have many advantages: the research problem is well defined, the study was conducted on a large sample of women, multivariate analysis methods were used, and a questionnaire is included. However, many issues need to be clarified. Difficult to accept are the tables.

First, the description of the tools and variables is unclear. There should be a clear definition of the dependent variable with the indicated question from the appendix. The abstract mentions a rate of 25.1%, which refers to non-participation in lifetime screening. The data shows that this is the percentage of non-participating from Tables 1-4. I suggest describing the dependent variable in the methods in the highlighted paragraph.

Also, the description of the last part of the questionnaire is not clear (lines 161+). It should include separate topic areas relating to the content of Tables 3 and 4 and the barriers described next, as these are the variables used in the analyses. Preferably with reference to the questions from the appendix.

Second, the tables contain a systematic error and many design flaws. This is probably due to copying the results from SPSS layout. The TOTAL column should be first, because it characterizes the sample, and here showing the percentages in the column is justified to show the structure). On the other hand, the columns now titled participants/non-participants should show percentages in the row. This follows the principle of showing the effect in the potential cause.  Then you can clearly compare each subgroup with the population prevalence of non-screening (25.1%).  This error is repeated in all tables.  As a result, some parts of the description do not match the data. For example: Women aged 30-49 showed higher screening participation compared to those aged ≤30 years. I expect 79% vs 71%, and the table shows the share of age groups among participants.

I also suggest changing some parts of the headline in all tables. This description participants/non-participants is not clear. You can give an overarching head participating in cervical cancer screening and below that no / yes.

Third, the description of the results focuses on the ORs from each table. I suggest after describing the demographic profile starting with the level of participation in screening. Currently, the key indicator of 25.1% is part of the description of this profile. Also missing is a commentary on the prevalence of certain beliefs or behaviours in the study group. It would be worthwhile to interpret the results from the Total column, to highlight particularly alarming data. These data are part of the discussion (line 349+) and there is no description of them in the results.

The discussion lacks a section on the limitations of self-study. Here one may have an objection to a lifetime perspective. It can be interpreted differently for very young and older women.  It would be advisable in the future to study participation according to national recommendations or for example last 1-2 year.

The Barriers to participating in cervical cancer screening paragraph from page 9 would be clearer if a chart was included with a brief description of what the main barriers were.

Editorial note: Under Table 4 not on analogous footnote as under previous tables.

Author Response

Comment 1: The study design and the method of presenting the results have many advantages: the research problem is well defined, the study was conducted on a large sample of women, multivariate analysis methods were used, and a questionnaire is included. However, many issues need to be clarified. Difficult to accept are the tables.
Response 1: We appreciate the reviewer's positive feedback regarding the research problem, sample size, analytical methods, and questionnaire inclusion. We acknowledge the concern about the need for clarification in several areas and the specific criticism of the tables. We have carefully re-examined the entire manuscript for clarity, paying particular attention to the Methods and Results sections. To address the concerns about the tables, we have comprehensively revised them to improve their clarity and readability. 

Comment 2: "First, the description of the tools and variables is unclear. There should be a clear definition of the dependent variable with the indicated question from the appendix. The abstract mentions a rate of 25.1%, which refers to non-participation in lifetime screening. The data shows that this is the percentage of non-participating from Tables 1-4. I suggest describing the dependent variable in the methods in the highlighted paragraph. Also, the description of the last part of the questionnaire is not clear (lines 161+). It should include separate topic areas relating to the content of Tables 3 and 4 and the barriers described next, as these are the variables used in the analyses. Preferably with reference to the questions from the appendix."
Response 2: We thank the reviewer for their insightful comments regarding variable definitions and questionnaire clarity. We have revised the Methods section, adding a paragraph explicitly defining 'lifetime non-participation in cervical cancer screening' as our dependent variable. This is a binary variable (yes/no) derived from question Q8_2 (Supplementary Material 1, Question Q8_2): 'Have you ever taken the Papanicolaou test?' 'No' responses indicate non-participation, explaining the 25.1% rate. We also substantially rewrote the questionnaire description, now providing a cohesive section detailing how the questions map onto the analyses presented in Tables 1-4 and the barriers analysis. Specifically, we describe questions related to awareness (Q6, Q9, Q11), healthcare engagement (Q7), perceptions and beliefs (Q10), and barriers (Q8_2_4), all referencing the Supplementary Material for precise question wording. Furthermore, we streamlined the Methods section to integrate the description of explanatory variables directly into the relevant questionnaire sections, eliminating redundancy and improving overall flow.

Comment 3: "Second, the tables contain a systematic error and many design flaws. This is probably due to copying the results from SPSS layout. The TOTAL column should be first, because it characterizes the sample, and here showing the percentages in the column is justified to show the structure). On the other hand, the columns now titled participants/non-participants should show percentages in the row. This follows the principle of showing the effect in the potential cause.  Then you can clearly compare each subgroup with the population prevalence of non-screening (25.1%).  This error is repeated in all tables.  As a result, some parts of the description do not match the data. For example: Women aged 30-49 showed higher screening participation compared to those aged ≤30 years. I expect 79% vs 71%, and the table shows the share of age groups among participants. I also suggest changing some parts of the headline in all tables. This description participants/non-participants is not clear. You can give an overarching head participating in cervical cancer screening and below that no / yes."
Response 3: We thank the reviewer for their meticulous and insightful critique of our tables. We have comprehensively revised Tables 1-4 to address the identified issues. The 'TOTAL' column is now presented first in each table, displaying the overall sample characteristics with column percentages. Critically, we have recalculated the percentages within the 'Participating in Cervical Cancer Screening' columns (now with subheadings 'No' and 'Yes') to represent row percentages. This change allows for direct comparison of non-participation rates within each characteristic subgroup to the overall non-participation rate, demonstrating the effect of each potential predictor. We have also updated the table headings for improved clarity, as suggested. Finally, we have carefully reviewed and revised the text describing the results in Tables 1-4 to ensure accurate interpretation and consistency with the revised table data. These changes significantly enhance the accuracy, clarity, and interpretability of our findings.

Comment 4: "Third, the description of the results focuses on the ORs from each table. I suggest after describing the demographic profile starting with the level of participation in screening. Currently, the key indicator of 25.1% is part of the description of this profile. Also missing is a commentary on the prevalence of certain beliefs or behaviours in the study group. It would be worthwhile to interpret the results from the Total column, to highlight particularly alarming data. These data are part of the discussion (line 349+) and there is no description of them in the results."
Response 4: We thank the reviewer for their insightful and valuable suggestions regarding the organization and interpretation of our results, particularly concerning the prominence of the overall participation rate and the inclusion of descriptive statistics alongside the association analyses. We agree that the original presentation could be improved to better highlight key epidemiological findings. Accordingly, we have made the following revisions to the 'Results' section:

  1. We have integrated key prevalence data from the 'Total' column of Tables 1-4 directly into the narrative of each existing sub-section (Demographic Profile, General Health/Lifestyle/Sexual Behavior, and Awareness/Engagement/Perceptions). This allows us to present the overall prevalence of a characteristic alongside its association with non-participation, providing a more complete and contextualized picture. For example, when discussing educational attainment, we now present both the overall distribution of education levels in the sample and the non-participation rates within each education level.
  2. Within the integrated descriptions, we have specifically drawn attention to findings that are particularly noteworthy or concerning from a public health perspective. This includes, but is not limited to, the low awareness of HPV (48.8%), the low awareness of the national screening program (51.0%), the relatively high proportion of women perceiving the Pap smear as expensive (22.2%), and the concerning prevalence of stigma-related avoidance behaviors (7.8%). These findings are now emphasized to underscore their importance.

Comment 5: "The discussion lacks a section on the limitations of self-study. Here one may have an objection to a lifetime perspective. It can be interpreted differently for very young and older women.  It would be advisable in the future to study participation according to national recommendations or for example last 1-2 year."

Response 5: We appreciate the reviewer's insightful comment regarding the limitations of self-reported data and the 'lifetime perspective' on screening history. We have incorporated this point into the newly added limitations paragraph in the Discussion section. We acknowledge that recall bias may affect the accuracy of reported screening history, and that the interpretation of 'never screened' differs significantly across age groups. We agree that future research should ideally focus on screening participation within specific timeframes, such as the nationally recommended intervals or the past 1-2 years, to provide a more precise and age-relevant assessment of screening behavior. This would allow for a more nuanced understanding of adherence to current guidelines.

Comment 6: "The Barriers to participating in cervical cancer screening paragraph from page 9 would be clearer if a chart was included with a brief description of what the main barriers were."

Response 6:  We agree that a visual representation of the barriers to cervical cancer screening enhances clarity and comprehension. We have added a new figure (Figure 1) to the 'Results' section, presenting a bar chart that displays the frequency of each self-reported barrier.

Comment 7: "Editorial note: Under Table 4 not on analogous footnote as under previous tables."

Response 7: We thank the reviewer for pointing out this oversight. We have added the missing footnote to Table 4 to ensure consistency with the formatting of Tables 1, 2, and 3. The footnote now includes explanations for the statistical significance symbols, abbreviations (CI, OR), the adjustment variables used in the multivariable analysis, and a note about potential discrepancies in the 'Total' column percentages due to rounding and/or missing values.

Reviewer 3 Report

Comments and Suggestions for Authors

Title: Understanding Cervical Cancer Screening Attendance: Barriers and Facilitators in Representative Population Survey

The cervical cancer one of the leading causes of cancer type among women. In the Europen countires the mortality of the cervix cancer is the  highest in Romania, therefore it is important to investigate what factors influencing the cervical cancer screening attendance.

Abstract: Please provide some exact health policy recommendations to improve the participation rate in the conclusion.

Introduction: Please provide a short description about the Romanian National Cervical Cancer Screening Program.

Methods: I do not understand clearly this sentence in the participation section: „This report focuses on the analysis conducted on a sub-sample of Romanian participants who reported that they had never been screened for cervical cancer.” In your tables it have been analysed the data of 1605 participants. Please clarify it.

Results: There is a duplication about the age avereag of the sample.

Please provide the main strenghts and limitations of the study.

Author Response

Comment 1: "The cervical cancer one of the leading causes of cancer type among women. In the Europen countires the mortality of the cervix cancer is the highest in Romania, therefore it is important to investigate what factors influencing the cervical cancer screening attendance."
Response 1: We appreciate the reviewer's comment highlighting the significance of cervical cancer as a public health issue, particularly in Romania. We agree that the high burden of disease in Romania underscores the importance of our study.

Comment 2: "Abstract: Please provide some exact health policy recommendations to improve the participation rate in the conclusion."
Response 2: We thank the reviewer for their suggestion to include specific health policy recommendations in the abstract's conclusion. We have revised the conclusion to be more actionable and directly address key findings. The abstract now explicitly states: "To improve participation, we recommend: enhanced physician referrals, HPV awareness campaigns, addressing the social stigma, and widespread communication about free screening availability." This concise statement provides concrete policy recommendations directly linked to the barriers and facilitators identified in our study, highlighting the practical implications of our research.

Comment 3: "Introduction: Please provide a short description about the Romanian National Cervical Cancer Screening Program."
Response 3: We have expanded the Introduction to include a concise description of the Romanian National Cervical Cancer Screening Program, directly addressing this comment. The revised section now details the program's launch in 2012, its provision of free Pap smear testing for women aged 25-64, and its aim to reduce cervical cancer incidence and mortality (Nyanchoka et al., 2022; Păvăleanu et al., 2018; National Health Insurance House, 2012). Furthermore, we have incorporated information about the program's low participation rates, particularly among vulnerable groups, and have situated this program within the broader context of national cervical cancer prevention efforts, including the reintroduction of HPV vaccination. This expanded description provides necessary background and context for our study.

Comment 4: "Methods: I do not understand clearly this sentence in the participation section: „This report focuses on the analysis conducted on a sub-sample of Romanian participants who reported that they had never been screened for cervical cancer.” In your tables it have been analysed the data of 1605 participants. Please clarify it."
Response 4: We thank the reviewer for highlighting this inconsistency. We have revised the sentence in the 'Participants' section to reflect the analysis conducted accurately. The sentence previously stated that the analysis focused on a sub-sample of unscreened women. This was incorrect. The analysis includes the selected sample of 1556 participants, comparing those who have never been screened for cervical cancer (the non-participants) to those who have been screened at least once (the participants) to identify factors associated with non-participation. The revised sentence now clarifies this, and we apologize for the confusion.

Comment 5: "Results: There is a duplication about the age avereag of the sample."
Response 5: We thank the reviewer for their careful attention to detail. The previous version of the manuscript inadvertently reported the mean age of the sample in duplicate. We have removed the redundant mention, and the mean age (46.0 years, SD = 10.63) is now reported only once in the 'Demographic Profile' section of the Results, ensuring clarity and avoiding repetition.

Round 2

Reviewer 2 Report

Comments and Suggestions for Authors

Perfect cooperation. I have no further comments.